# Switching Mediterranean Consumers to Mediterranean Sustainable Healthy Dietary Patterns (SWITCHtoHEALTHY): Study Protocol of a Multicentric and Multi-Cultural Family-Based Nutritional Intervention Study

**DOI:** 10.3390/nu16223938

**Published:** 2024-11-18

**Authors:** Lorena Calderón-Pérez, Alícia Domingo, Josep M. del Bas, Biotza Gutiérrez, Anna Crescenti, Djamel Rahmani, Amèlia Sarroca, José Maria Gil, Kenza Goumeida, Tianyu Zhang Jin, Metin Güldaş, Çağla Erdoğan Demir, Asmaa El Hamdouchi, Lazaros P. Gymnopoulos, Kosmas Dimitropoulos, Perla Degli Innocenti, Alice Rosi, Francesca Scazzina, Eva Petri, Leyre Urtasun, Giuseppe Salvio, Marco de la Feld, Noemi Boqué

**Affiliations:** 1Eurecat, Technology Centre of Catalonia, Nutrition and Health Unit, 43204 Reus, Spain; lorena.calderon@eurecat.org (L.C.-P.); alicia.domingo@eurecat.org (A.D.); anna.crescenti@eurecat.org (A.C.); 2Eurecat, Technology Centre of Catalonia, Biotechnology Area, 43204 Reus, Spain; josep.delbas@eurecat.org (J.M.d.B.); biotza.gutierrez@eurecat.org (B.G.); 3Centre de Recerca en Economia i Desenvolupament Agroalimentari, Universitat Politècnica de Catalunya, 08860 Barcelona, Spain; djamel.rahmani@upc.edu (D.R.); amelia.sarroca@upc.edu (A.S.); chema.gil@upc.edu (J.M.G.); kenza.goumeida@upc.edu (K.G.); tianyu.zhang@upc.edu (T.Z.J.); 4Nutrition and Dietetics Department, Faculty of Health Sciences, Görükle Campus, Bursa Uludag University, 16059 Nilufer-Bursa, Turkey; mguldas@uludag.edu.tr; 5Biotechnology Department, Graduate School of Natural and Applied Sciences, Görükle Campus, Bursa Uludag University, 16285 Bursa, Turkey; cagla.erdgn17@gmail.com; 6Unité de Recherche Nutrition et Alimentation, National Center for Nuclear Energy, Science and Technology CNESTEN, Rabat-Kenitra 14000, Morocco; asmaaelhamdouchi@gmail.com; 7Information Technologies Institute, Centre for Research and Technology Hellas, GR 570 01 Thermi, Greece; lazg@iti.gr (L.P.G.); dimitrop@iti.gr (K.D.); 8Human Nutrition Unit, Department of Food and Drug, University of Parma, 43125 Parma, Italyalice.rosi@unipr.it (A.R.); francesca.scazzina@unipr.it (F.S.); 9National Centre for Food Technology and Safety, 31570 San Adrian, Spain; epetri@cnta.es (E.P.); lurtasun@cnta.es (L.U.); 10ENCO Consulting, 80122 Naples, Italy; salvio@enco-consulting.it (G.S.); m.delafeld@enco-consulting.it (M.d.l.F.)

**Keywords:** Mediterranean diet, multicentric study, family intervention, educational materials, sustainability, digital tool, educational game, healthy snacks

## Abstract

Background/Objectives: Populations in Mediterranean countries are abandoning the traditional Mediterranean diet (MD) and lifestyle, shifting towards unhealthier habits due to profound cultural and socioeconomic changes. The SWITCHtoHEALTHY project aims to demonstrate the effectiveness of a multi-component nutritional intervention to improve the adherence of families to the MD in three Mediterranean countries, thus prompting a dietary behavior change. Methods: A parallel, randomized, single-blinded, and controlled multicentric nutritional intervention study will be conducted over 3 months in 480 families with children and adolescents aged 3–17 years from Spain, Morocco, and Turkey. The multi-component intervention will combine digital interactive tools, hands-on educational materials, and easy-to-eat healthy snacks developed for this study. Through the developed SWITCHtoHEALTHY app, families will receive personalized weekly meal plans, which also consider what children eat at school. The engagement of all family members will be prompted by using a life simulation game. In addition, a set of activities and educational materials for adolescents based on a learning-through-playing approach will be codesigned. Innovative and sustainable plant-based snacks will be developed and introduced into the children’s dietary plan as healthy alternatives for between meals. By using a full-factorial design, families will be randomized into eight groups (one control and seven interventions) to test the independent and combined effects of each component (application and/or educational materials and/or snacks). The impact of the intervention on diet quality, economy, and the environment, as well as on classical anthropometric parameters and vital signs, will be assessed in three different visits. The COM-B behavioral model will be used to assess essential factors driving the behavior change. The main outcome will be adherence to the MD assessed through MEDAS in adults and KIDMED in children and adolescents. Conclusions: SWITCHtoHEALTHY will provide new insights into the use of sustained models for inducing dietary and lifestyle behavior changes in the family setting. It will facilitate generating, boosting, and maintaining the switch to a healthier MD dietary pattern across the Mediterranean area. Registered Trial, National Institutes of Health, ClinicalTrials.gov (NCT06057324).

## 1. Introduction

Currently, unhealthy diets are recognized as one of the most important risk factors in the development of noncommunicable diseases (NCDs). According to the Global Burden of Disease study (GBD) 1990–2019 [1,2], the consumption of unhealthy foods and nutrients, including sugar-sweetened beverages, red meat, and sodium, is far higher than the optimal intake in young and middle-aged adults. Diet modernization has led to a shift from traditional dietary patterns to more convenient, processed, and less nutritious food choices, with negative health consequences like overweight, obesity, diabetes, elevated blood pressure, and hyperlipidemia, all metabolic risk factors for NCDs [3]. In 2023, more than 150 million people suffered from NCDs in the Eastern Mediterranean region, with cardiovascular diseases (CVDs) accounting for 17.9 million deaths [3]. These figures corroborate the increasingly evident abandonment of the Mediterranean traditional dietary pattern [4].

The Mediterranean diet (MD) is linked to a set of skills, knowledge, practices, and traditions ranging from cultivation to the processing, preparation, and particularly the consumption of food and defines a unique lifestyle recognized as a common cultural heritage of Mediterranean communities [5]. It is essentially a plant-based dietary pattern based on the consumption of high amounts of fresh fruits, vegetables, and legumes as the main sources of fiber and antioxidant compounds and cereals (mainly whole grain), nuts, and olive oil as the main sources of fat. It also includes an abundance of fish and shellfish and a moderate consumption of white meat, eggs, and dairy products (mainly yogurt and cheese) [6,7]. The MD is a sustainable food model that is recognized as (a) environmentally sustainable, protecting biodiversity; (b) culturally acceptable and fair; (c) economically accessible and affordable; and (d) nutritionally adequate, safe, and healthy, as represented in the new environmental dimension of the MD pyramid [7,8]. According to the results of a recent meta-review, higher MD adherence scores were significantly associated with a reduced risk of CVD, type 2 diabetes, overall cancer mortality, different cognitive diseases, fractures, and metabolic syndrome, in addition to lower body weight, BMI, and blood pressure [9]. The available literature regarding MD adherence in children and adolescents from Mediterranean countries shows a higher percentage of young people with low MD adherence than with high adherence, according to the KIDMED index [10,11]. These findings suggest that several factors are contributing to the erosion of traditional Mediterranean dietary habits, among which cultural, socioeconomic, and lifestyle changes have a primary role [12]. The abandonment of traditional eating habits, the high consumption of precooked food, the low consumption of fresh, local, and seasonal products, and a sedentary lifestyle denote profound social changes in the current food culture of the Mediterranean region, especially among younger generations [4,12]. All of this is aggravated by the decline in agricultural biodiversity, largely due to globalizing trends, climate change, and the mechanization of farming practices, which reduce the sustainability of local production systems and, with it, our ability to safeguard the Mediterranean diet [13].

Nevertheless, parental factors need to be considered, especially when it comes to barriers and facilitators for children’s dietary patterns. The family food environment becomes a strong predictor of children’s food preferences and eating behaviors. Parental food habits and feeding strategies are the most dominant determinants of a child’s food choices, which will persist throughout their life [14]. Some preliminary studies conducted with children and adolescents have found a significant positive correlation between MD adherence and family socioeconomic status, revealing that a higher MD adherence was associated with a better mother’s education and income level [11]. In addition, in the PASOS study recently conducted in 3607 children and adolescents in Spain, adherence to the MD was higher in children and adolescents with healthier parental lifestyle [15].

It is believed that the social context most likely to induce healthy behavior changes is the family [16]. The reciprocal adult–child relationship represents a way of influencing the behavior of both children and adults. Lasting dietary changes are more likely when they involve the family unit because of the increased likelihood that family members will act and sustain behaviors and because of the beneficial familial social support. However, the current evidence of interventions aimed at improving dietary habits by involving all the family is limited [17,18]. Most of these interventions are focused on the prevention or treatment of child obesity, evaluating the effectiveness of improving diet quality at home on classical health variables such as body composition, glucose, and lipid profiles [19,20,21]. But, as far as we know, there is no family-based intervention aimed at improving the adherence to MD. Although many interventions carried out in schools have been shown to be effective in promoting a healthier dietary pattern, it is not fully determined if these interventions result in changed behavior outside the school setting [17,22]. In fact, an intervention comparing a family-based, school-based, and combined approach demonstrated that changes in parenting practices and healthy dietary behaviors were achieved among those exposed to the family intervention [23].

Recent trends show an emerging number of technology-based or eHealth interventions to improve health outcomes for lifestyle behaviors, particularly in young populations [24,25,26,27]. However, there is poor evidence of the effectiveness of family-based nutrition education programs engaging parents and/or parents and their children directly using digital tools such as mobile apps [28]. Most of the existing mobile apps promoting the MD lack family-based dietary guidance in terms of food preferences, sociocultural aspects, and other family-related parameters. Moreover, there is a need for complementary nutrition interventions to build a supportive environment for effective healthy eating in the household [28,29,30]. Codifying the pros and cons of using digital tools in health/dietary interventions, we note as benefits the possibility for personalized and adaptive guidance, the interactive engagement of all family members, the facilitation of easy, consistent, and ongoing support, the encouragement of self-monitoring and reflection, and the facilitation of education through experimental learning; and, as drawbacks, we mention digital divide and accessibility issues, challenges related to sustained engagement over time, family digital dynamics and differences in motivation to use digital tools, complexity and usability issues, and, finally, challenges related to privacy and data security.

In this context, the SWITCHtoHEALTHY project aims to generate a dietary behavior change by demonstrating and reinforcing the role of the family in promoting a sustainable change towards enhancing adherence to the Mediterranean dietary pattern of all the family members (adults, adolescents, and children). To achieve this goal, a randomized, single-blinded, and controlled multicentric nutritional intervention study, involving families from three Mediterranean countries (Spain, Turkey, and Morocco), will be carried out. The study will test the effectiveness of a holistic family-based approach that combines. (1) digital interactive tools (SWITCHtoHEALTHY app) designed to support parents in preparing weekly healthier dietary plans for all the family members; (2) hands-on educational materials and activities designed through a learning-through-playing approach to increase adolescents’ knowledge and motivation on healthy and sustainable food habits; and (3) healthy and sustainable plant-based snacks to be introduced in children’s dietary plans to substitute less healthy options between meals. The plant-based snacks, digital tools (app and game), and educational materials will be developed based on their effectiveness in promoting adherence to the MD, addressing common barriers Mediterranean families face in adopting healthier dietary habits.

The intervention is grounded in the COM-B model, part of the Behavior Change Wheel (BCW) framework, focusing on capability, opportunity, and motivation as key drivers of behavior change. It integrates Social Cognitive Theory (SCT) principles, emphasizing family-based support and digital engagement through tools. This combination aims to support MD adherence by targeting individual knowledge, family reinforcement, and accessible resources [31].

A sustained enabling environment will be promoted throughout the project to increase the awareness of an MD-based healthy diet and lifestyle model across the Mediterranean region in eight participating countries (SWITCHtoHEALTHY project; https://switchtohealthy.eu/ (accessed on 3 June 2024)), namely Spain, Italy, Greece, Turkey, Lebanon, Egypt, Tunisia, and Morocco, involving different research institutions with complementary expertise and resources within the areas of nutrition, eHealth, digital technologies/ICT, agro-food, food eco-design, innovation, R&D&I, and food manufacturing.

## 2. Materials and Methods

### 2.1. Study Design and Setting

The SWITCHtoHEALTHY study is a parallel, randomized, single-blinded, and controlled multicentric nutritional intervention study. It aims to evaluate the effects of a multi-component intervention deployed at the family level, combining digital interactive tools and hands-on educational materials and activities with healthy and sustainable plant-based snacks, on the adherence to the MD pattern in 480 families with adolescents and children from three Mediterranean countries (Spain, Turkey, and Morocco), with 160 families per country. Secondarily, the effects of the multi-component intervention will be assessed according to lifestyle, anthropometric, dietary, sociodemographic, socioeconomic, and environmental indicators.

This study protocol was registered at ClinicalTrials.gov (NCT06057324) and is in accordance with the Helsinki Declaration and the Good Clinical Practice Guidelines on the International Conference of Harmonization. All the procedures described in the protocol, informed consent, and other documents have been reviewed and approved by the Ethics Committees of the three centers involved in the study: (1) Fundació Eurecat (study co-ordinator): Institut d’Investigació Sanitària Pere Virgili, IISPV (Ref. 135/2023); (2) Ethics Committee of Bursa Uludag University Faculty of Medicine Clinical Researches (Ref. 2023-23/29); and (3) Comité d’Ethique pour la Recherche Biomédicale (CERB), Université Mohammed V Faculté de Médecine et de Pharmacie de Rabat (preapproval). This protocol complies with the Standard Protocol Items: Recommendations for Clinical Trials (SPIRIT 2013 Statement and SPIRIT-AI extension) [32,33].

The 160 families recruited in each country will be randomly assigned into eight study groups (*n* = 20 families per group per country) according to a multi-component intervention including different combinations of the three study components (artificial intelligence (AI)-based app for all family; educational materials and empowerment activities for adolescents; and healthy plant-based snacks for children) and a control group that will receive basic Mediterranean guidelines for parents and their children (Figure 1).

To evaluate the overall effect of the intervention as well as the individual and combining effect of each intervention component, a full-factorial design will be used [34]. This design will allow us to evaluate the independent effect of the different intervention components versus the control group and, simultaneously, the summatory effect of combining different components on increasing the family adherence to the MD. This will therefore provide a stronger inference about the effect on the different implemented interventions [35].

The overall study duration will be 13 months for each participating family, including 3 main visits (V0, V1, and V2) (Figure 2). A prior preassessment visit will be performed at baseline, followed by the intervention study that will be deployed in the last 3 months (between month 10 and 13). According to this timeframe, this study will be arranged in two different phases (Figure 2).

*Phase 1* (*month 0 to month 9*): preassessment. The families will be recruited and included in this study; they will be randomly assigned to the different study groups; and consumption patterns and sociodemographic, socioeconomic, and environmental indicators will be evaluated through specific questionnaires. In addition, the different educational materials and activities, AI-based app, and healthy plant-based snacks will be developed prior to the start of the intervention study. This information will help us understand the context of each participant group and inform our analysis.

*Phase 2* (*month 10 to month 13*): the intervention study will be deployed at the family level. Families will receive the educational materials and activities, the AI-based app, and the plant-based snacks developed in Phase 1, depending on their assigned intervention group. In addition, anthropometrics, lifestyle, dietary, and socioeconomic and environmental indicators will be evaluated through specific questionnaires.

### 2.2. Family Recruitment and Eligibility

The Spanish families will be recruited by researchers from the Nutrition and Health Unit (Eurecat, Reus) and the Center for Research in Agri-Food Economics and Development (CREDA, Barcelona), and the Turkish and Moroccan families will be recruited by researchers from the Bursa Uludag University and the Unité de Nutrition et al.imentation CNESTEN, respectively. In addition, all countries will have the support of schools, high schools, family associations, and other collaborators such as primary care centers and community centers.

Families will be eligible to participate if they meet the following criteria: (a) families from any socioeconomic status with at least one child younger than 12 years and one adolescent older than or equal to 12 years (overall age range 3–17 years) and that live together; (b) informed consent has been signed by parents and the adolescent; and (c) families have a mobile phone, tablet, or computer with internet access. This inclusion criterion of composition of families and ages of children was selected to capture a range of developmental stages within families, as younger children and adolescents may respond differently to dietary interventions than adults. While this may introduce differences in family structure, it also promotes interaction across ages, potentially enhancing the adoption of healthy eating habits through sibling influence and parental support.

Exclusion criteria will be as follows: (a) having allergies or food intolerances to any of the following plant-based snack ingredients: fruit, vegetables, legumes, cereals, seeds, nuts, and yogurt (only for children under 12 years); (b) disliking any of the snack products (only for children under 12 years); (c) MD Adherence Score (MEDAS) or KIDMED score greater than or equal to 11 points, which is a food pattern already highly concordant with the MD [36]; (d) following a vegan diet (any of the family members); (e) following a prescribed long-term and strict diet for any reason, including diets for weight loss and diets for chronic metabolic or autoimmune disorders such as type 1 diabetes, celiac disease, inflammatory bowel disease, or rheumatoid arthritis (any of the family members); (f) participating in or having participated in a clinical trial or nutritional intervention study in the last 30 days prior to inclusion in this study; (g) no or limited access to the internet; and (h) being unable to follow the study guidelines.

All recruited families will be interviewed by expert nutritionists to assess whether they meet the conditions for inclusion. For this purpose, two principal researchers in each country will be responsible for explaining the background, purpose, process, risks, and benefits of this study and obtaining informed consent.

### 2.3. Interventions

The interventions will involve a combination of digital-based tools and hands-on educational materials and activities, with healthy plant-based snacks developed during Phase 1 of this study. At the beginning of Phase 2 (month 10), there will be prior training for all families, where they will be informed about the specific intervention to which they will be randomly assigned.

The intervention materials and products will be adapted to reflect the cultural preferences and dietary habits of each country. This approach ensures that the interventions are relevant and engaging for families in each country. However, we will maintain some common elements across all interventions to facilitate meaningful comparisons in the study’s outcomes.

#### 2.3.1. Digital Interactive Tools

The digital interactive tools will be implemented in the form of an integrated web application, the “S2H app”, which will be used by parents to empower them in their daily family dietary choices. The S2H app will be based on the integration of two separate dietary apps and an educational dietary game that will be designed, developed, technically validated, and extensively tested with real users during Phase 1 of the study: AI-based app for parents, app for children’s Med-based dietary plan, and Digitami Game (Table 1). The two constituent dietary apps are, in turn, based on pre-existing technologies that have been validated in previous research projects with real users: the EU-funded PROTEIN project https://cordis.europa.eu/project/id/817732 (accessed on 9 September 2024) and the NUBI Parma app https://www.ailab.unipr.it/projects/ (accessed on 9 September 2024). Otherwise, the educational game will be developed entirely within the SWITCHtoHEALTHY project.

The S2H app engine is fed with (a) expert-approved dietary rules; (b) user profile data for all family members; (c) school menus (from the school catering system) or, alternatively, lunch box contents and school café proposals; and (d) a list of representative meals of the local Mediterranean cuisine, i.e., breakfasts, lunches, and dinners, along with caloric and macronutrient information (the latter for the parents only). The seasonality of each food and ingredient and the culinary traditions of the food area in which the app is used are also considered. All the contents in the S2H app will be available in 4 languages (English, Spanish, Turkish, and Arabic).

The users will be able to provide basic profile information for each family member, including their year of birth, sex, physical characteristics (weight, height, and BMI), physical activity levels, dietary preferences, and history of food allergies, and will then receive weekly meal plans for them, adhering to a healthy and sustainable MD. Conversely, more general information will be requested for child users regarding age group, allergy to milk proteins, consumption of the project snacks, and school name (for school menu) or lunch box and school café alternatives.

#### 2.3.2. Hands-On Educational Materials and Activities

The hands-on educational materials will be developed through a learning-through-playing approach to increase adolescents’ knowledge and motivation on healthy and sustainable food habits according to sustainable MD patterns. An MD toolkit will be provided to adolescents and their families to educate them on the MD and associated health benefits to encourage the consumption of more healthy foods. The materials of the MD toolkit will be designed during Phase 1 of this study, including the following: (a) specific games and activities for adolescents, with proposed activities played at home with parents, brothers and sisters, and friends; (b) a range of resources, including educational fact sheets combining more theoretical notions about the MD model and optimal diet management (e.g., the “healthy eating plate”, consumption frequencies, and food portions for different ages), infographics with take-home messages, recipes, and a short video. The educational materials will be available in different languages through the SWITCHtoHEALTHY website https://switchtohealthy.eu/ (accessed on 9 September 2024).

The educational activities will be part of an educational program specially designed to improve adherence to the MD of adolescents. The program will be conducted outside of school hours during the 3-month intervention and will be previously designed during Phase 1 of this study under the umbrella of three selected topics: (1) Mediterranean Diet and Lifestyle: discovering cultures, foods, and tastes; (2) Mediterranean Diet: from a healthy diet to a sustainable dietary pattern; and (3) Food Lab: kitchens from research or cooking centers as experimental hands-on zones for the application of healthy and sustainable dietary patterns. At least two cocreation sessions will be organized in each study country, with adolescents assigned to the educational activity groups as cocreators. The World Cafe Methodology [37,38,39] will be applied, encouraging participants to talk to each other and discuss the three selected topics in small groups in a cozy café-style space hosted by researchers as moderators (Ms) and high-school teachers as facilitators (Fs). Each session will take around one-and-a-half hours and will be joined by about 12 adolescents and at least 2 teachers and 2 researchers who will be neutral and encourage participation. The adolescents will be divided into three tables (4 to 5 adolescents per table) and, at each table, one of the three selected topics will be discussed so that the adolescents will rotate around every 20 min to participate in the discussion of all the topics (Figure 3). The researchers will record all the conversations that take place on a smartphone and will also take written notes. Once all the discussion rounds have finished and after a coffee break, the entire group will meet again in plenary. The resulting activities from each table will be summarized by researchers who will bring down the number of generated ideas to a handleable minimum, focusing on those factors generally considered to be the most important. Finally, the participants will vote on the results to prioritize actions to be implemented as educational activities in the frame of the educational program. After the cocreation session, all the data will be gathered and analyzed by the researchers in each country. The codesigned activities will then be reprioritized by all participants via an online survey. Lastly, the researchers will select and develop the 12 most voted educational activities, with 4 per topic, to be implemented within the framework of the intervention study.

During Phase 2 (month 10 to month 13), the codesigned activities will be carried out by adolescents at home (or in other nearby places, like markets) together with family and at research facilities together with the research staff. Four activities will be carried out on each selected topic in each month of intervention. Adolescents will be provided with a practical support handbook (or e-book) collecting all the activities and needed materials, which will be available and downloadable in different languages from the study website. Each activity will be described in the form of a didactic programing unit structured with the following sections: title, purpose, learning outcomes, preparation time and duration, methodology, materials required, and other resources. Due to the nature of this study, to avoid confounders in different intervention groups, these activities cannot be performed at school during the intervention. Nevertheless, the activity practical support handbook will be available for schoolteachers at the end of the intervention.

#### 2.3.3. Healthy Plant-Based Snacks

The healthy and sustainable plant-based snacks will be developed and produced during Phase 1 of this study by three food companies located in Spain (DELAFRUIT S.L.U., Tarragona), Morocco (ECOMAB SARL, Casablanca), and Turkey (KOCAHAN SEKERLEME, Bursa) and will be consumed during the 3-month intervention by children aged 3 to 11 years.

The composition of the new food products will be based on local and traditional vegetables, fruits, legumes, seeds, and nuts as main ingredients, which are rich in health-promoting bioactive compounds. Processing and preservation strategies will be adapted to retain the nutritional value of the ingredients, and an eco-design methodology will be followed to produce the snacks [40]. In addition, the selection of the ingredients will be carried out beforehand in the three study countries through participatory focus groups involving parents and children. The locally produced plant-based ingredients will be presented to the families to determine their level of knowledge of them and the degree of incorporation in their diets (Table 2). The results of the focus group will allow the food manufacturers to reach a better understanding of the fruits, nuts, and vegetables the children usually consume in each country. Finally, after the preindustrial scale-up, four different snack prototypes to be used during the intervention in each country will be chosen according to the family’s preferences, and the portions and frequency of consumption will be defined to complement the children’s mid-morning and mid-afternoon snack, considering their nutritional and energy needs.

Since the snacks are produced with healthy and Mediterranean ingredients free of toxic agents and contaminants, no problems relating to children’s safety are expected during the intervention study. At the time of family selection, the children’s allergy history will be checked and the organoleptic characteristics of the snack products will be described to parents to verify the children’s acceptance.

The snack products will be produced in a single batch in each country. Once produced, the manufacturers of each country will send them to the corresponding enrolment center prior to the start of the intervention study. The products will be delivered in a lot to the families in the same country at the beginning of the intervention.

### 2.4. Visits and Procedures

All families will attend the school facilities or at research centers where they are recruited on 3 different occasions (V0, V1, and V2). The visits will be divided between the two study phases previously described. Figure 4 shows the schedule of each visit, which will follow the same timeline for each enrollment center.

In Phase 1, families will be recruited and a preassessment visit will be carried out (V0). At this point, families will be assessed against the eligibility criteria and then informed about this study in a group session. The informed consent document will be signed by parents and adolescents. The researchers will then instruct them on the completion of online questionnaires at home through the Microsoft Forms web-based application to assess socioeconomic, sociodemographic, lifestyle, and environmental indicators. This information will be valuable in predicting selected behavior sub-domains that will be evaluated based on a theoretical model derived from the COM-B model for behavior change. A theoretical model based on information on physical activity, food habits, lifestyle, and social behavior will be created, aiming to achieve the best outcomes via surveys that are not too long [41]. In addition, families will complete a 7-day semi-weighted food record (7DWR) to assess dietary habits and consumption patterns at baseline. After V0, the selected families will be randomly allocated to the different intervention/control groups.

In Phase 2, the selected families will attend V1 and V2 at the beginning and end of the 3-month intervention study, respectively. In V1, families will receive group training according to the different intervention groups: on the use of the S2H app, on snack consumption, and/or on the use of educational materials and activities. In addition, each family will be visited individually by expert nutritionists who will assess adherence to the MD through the MEDAS and KIDMED questionnaires for adults and children, respectively, and will take anthropometric measurements (body weight, height, BMI, waist circumference, blood pressure, and heart rate) for each family member. Nutritionists will also complete a medical history, including the use of medications and/or supplements. At the end of V1, time will be left for families to self-complete online questionnaires through the Microsoft Forms web-based application. The questionnaires will be adapted according to the age of the family members to assess socioeconomic, lifestyle, dietary, and environmental indicators. In addition, families will complete a 7DWR and report food expenses during the week after V1.

During the 3-month intervention, families will use the different educational components and products. They will be in continuous contact with the research team to ensure compliance and report any adverse events, particularly in children assigned to consume the snack products. In V2, each family will be visited individually, the same parameters as those in V1 will be measured by nutritionists, and data on socioeconomic, lifestyle, dietary, and environmental indicators will be self-reported by each family member by means of online questionnaires. A closing session of this study will be carried out by researchers once the intervention is completed to inform the families of the main results of the study.

### 2.5. Outcomes

The primary outcome will be adherence to the MD measured through the MEDAS and the KIDMED questionnaire score in parents and children, respectively. The MEDAS is an extension of a 9-point score developed in the Prevención con Dieta Mediterranea (PREDIMED) trial [42,43]. It consists of 2 questions about eating habits, 8 questions about the frequency of consumption of typical foods of the MD, and 4 questions about the consumption of foods not recommended in this diet. Each question is scored with 0 (noncompliant) or 1 (compliant), and the total score ranges from 0 to 14, so a score of 14 points means maximum adherence. The KIDMED score consists of 16 items with affirmative or negative answers, where there are 4 questions denoting a negative connotation to the MD and 12 questions denoting a positive connotation. A positive answer to questions denoting negative connotation is scored with −1, while positive connotation questions are scored with +1. The total score ranges from 0 to 12, so a score of 12 points indicates good adherence [44]. To avoid deviations in outcomes (such as physical activity or portions of food), surveys will be assessed by researchers during visits.

The secondary outcomes will include the following:Anthropometric data: body weight, measured in all family members with a Tanita DC 430S-MA (Tanita Corp., Barcelona, Spain); height, obtained using a wall-mounted stadiometer (Tanita Leicester Portable; Tanita Corp., Barcelona, Spain); waist circumference, measured at the level of the narrowest region between the last costal (10th rib) and the edge of the iliac crest using a 200 cm anthropometric steel measuring tape; and BMI, obtained using the formula of weight in kilograms divided by the square of the height in meters.Systolic and diastolic blood pressure and resting heart rate, measured twice at 1 min intervals using an automatic sphygmomanometer (OMRON HEM-907; Peroxfarma, Barcelona, Spain).Sociodemographic and socioeconomic factors, measured in parents and adolescents via an adapted questionnaire from Rodríguez-Rodríguez et al. [45]. This questionnaire will collect information regarding age, gender, household members, marital status, income, level of education, and employment status.Social factors, measured in all family members via an adapted questionnaire from McIntosh et al. [46]. This questionnaire will collect information regarding work flexibility and satisfaction, work–life balance, lifestyle, time spent with children, family decision making, and task responsibility.Physical activity, measured in all family members by validated physical activity questionnaires, IPAQ for adults [47] and PAQ-C [48] and PAQ-A [49] for children aged over 8 years and adolescents, respectively.Health factors, measured in all family members via the General Health Questionnaire (GHQ) [50]. This questionnaire will ask about different aspects related to health and healthy lifestyle such as sleeping habits, hospitalization, and prescribed medications.Mediterranean lifestyle habits, measured only in parents via the Mediterranean Lifestyle index (MEDLIFE), which will capture adherence to an overall Mediterranean healthy lifestyle [51].Diet quality, measured via the Healthy Diet Index (HDI) [52] for adults and KIDMED [44] for adolescents and children.Quality of life, measured via the EQ-5D-5L scale for adults [53] and the KIDSCREEN scale for adolescents and children aged over 8 years [54]. The EQ-5D-5L assesses five dimensions: mobility, self-care, usual activities, pain/discomfort, and anxiety/depression. The KIDSCREEN assesses children’s and adolescents’ subjective health and well-being.Nutritional knowledge, measured via a short consumer-oriented nutrition knowledge questionnaire for adults [55] and the HELENA questionnaire for adolescents, which also includes a self-reported food frequency questionnaire [56]. These questionnaires will measure the parents’ and adolescents’ knowledge of a healthy diet.Attitudes and beliefs in relation to food, measured via an adaptation of the McIntosh et al. questionnaire [46] and the Early Parenting Attitudes Questionnaire (EPAQ) [57]. These questionnaires will assess the family eating habits and attitudes of and barriers faced by parents and adolescents in relation to the MD.Family weekly food intake, measured in all family members via a 7DWR, which involves an individual weighing of each food item prior to consumption. The 7DWR will be completed during the week after visit V0 and V1 and the week prior to visit V2. In V0, families will receive instructions for the correct completion. Nutritionists will instruct families on the use of kitchen scales. Apart from food, 7DWR will allow data on the average intake of macro- and micronutrients to be obtained.Family weekly food expenses, measured per family unit using shopping receipts collected during the week after visit V0 and V1 and in the week prior to visit V2. In each visit, families will be provided with an envelope for collecting the shopping receipts and other food expenses for a week.Environmental impact of food consumption, assessed by converting the 7DWR consumption quantities into kg of CO_2_ emission, land use, and water footprint using existing data in the literature on the environmental impacts of different food groups [58,59,60].Economic impact, estimated using the data obtained from family weekly food records and expenses reported by means of shopping receipts. Two measurements will be carried out at the beginning and end of the 3-month intervention, and potential effects of promotions at the specific retail outlet as well as family events and the seasonal effect will be controlled.

Each specific outcome and visit are summarized in Figure 4.

The integrated data from the primary and secondary outcomes, including MD adherence, diet quality, family weekly food intake, and food expenses, will provide an overview of the three main impact dimensions: quality of diet and health, economic, and environmental. These indicators will allow us to better understand the reasons behind dietary behavioral changes to facilitate the scaling-up of the most effective interventions or to redesign the less effective ones in the future.

### 2.6. Randomization and Allocation

After the baseline preassessment visit (V0), participating families will be randomized into the eight study groups. The randomization sequence list will be generated using the Randomization.com website (http://www.randomization.com) (accessed on 13 September 2024) by an external researcher not involved in this study. Stratified randomization, employing blocking, will be conducted to achieve a 1:1:1:1:1:1:1:1 ratio for the 7 intervention groups and the control group; *n* = 60 families per block. Due to the nature of the study, the families and the research staff cannot be blinded to the intervention, although the investigators who will perform the data analysis will work under concealment of the assignment.

### 2.7. Sample Size

The sample size was calculated based on differences in MD adherence score (primary outcome) measured by MEDAS and KIDMED questionnaires. The sample size was estimated using the analysis of variance (differences between 8 study groups). It was assumed that the alpha value would be equal to 0.05 and beta 0.20, exactly as reported in previous studies [61], with a common standard deviation equal to 1.19. The minimum group size needed to detect a minimum increase of 1 point in the intervention group’s MEDAS compared to the control group is equal to 45 families per study group (15 families per country). The total sample is, therefore, 45 × 8 = 360 families. However, assuming a 25% drop-out rate, the minimum group size is equal to 60 families per study group (20 families per country). Overall, 160 families will thus be recruited per country after obtaining the approval of the Ethics Committee of each recruiting center and written informed consent from parents and adolescents.

### 2.8. Criteria for Discontinuing or Modifying Allocated Intervention

If a severe or unanticipated adverse event (AE) occurs that may influence the risk/benefit ratio of this study, the principal investigator must report this to the ethics committee and permanently discontinue the family from the intervention. Participating families will be asked to report any AEs experienced during the entire intervention. For each AE, the onset date, intensity, relationship to the study product, action taken, and outcome will be recorded by researchers in accordance with the Medical Dictionary for Regulatory Activities (MedDra dictionary), Version 24.0 of MedDRA (Spanish), March 2021. In case of the suspicion of a major AE, such as an allergic reaction, the case will be referred directly to the study doctor, who will evaluate the symptoms and signs and adopt the necessary measures according to their severity.

In addition, the researchers can withdraw a family from this study if they consider that the family can no longer meet all the requirements of the study or if any of the procedures are considered possibly harmful for any of its members. Families participating in the study will be withdrawn if they meet any of the following criteria: (1) any circumstance that prevents them from performing the study procedures (i.e., using the educational materials or digital interactive tools and complying with the intake of the study products); (2) not attending study visits.

The losses and dropouts will be carefully recorded to ensure study reliability. The next participant who meets the eligibility criteria will replace the last participant who withdraws from the research until reaching the calculated sample size.

### 2.9. Data Management

All data will be collected and evaluated by experienced researchers and registered by the same researchers or study participants through predesigned forms in the Microsoft Forms web-based application.

The online forms will be self-completed by families at home in V0 and at the research center or school facilities in V1 and V2 with the assistance of the research staff. The forms will be adapted from validated scales for adults, adolescents, and children, and translated into the language of each country (Spanish, Turkish, French, and Arabic). They will comprise nine blocks of questions assessing various factors of interest (attitudes, intentions, self-efficacy, time allocation, work and household stress, health, nutrition, etc.), all of them integrated using a theoretical framework matrix [46,55,62] (Table 3).

Each family will be provided access to the Microsoft Forms links with a unique code previously assigned. Each member of the family must complete their own forms, except for children under 8 years of age, who will need the help of their parents for completion. In total, parents must complete 3 forms, adolescents 2 forms, and children under 12 years of age only 1 form. The average time for parents to complete the forms is 40 min, while it will only take about 20 min for adolescents and children under 12 years.

Anthropometric measurements will be collected at V1 and V2 by trained nutritionists with specific devices and registered in an online data collection notebook available for each family member in the Microsoft Forms web-based application.

Data on consumption patterns will be recorded in writing format in a notebook including a 7DWR. For one week, families must record all the foods and drinks and the amount consumed for each member of the family. In addition, they will have to record food expenses and keep shopping receipts. The researchers will check the data provided and transfer them to a common Excel database for the three enrolled countries to assess the economic and environmental impact.

Data generated with Microsoft Forms will be exported to Microsoft Excel. Access to the database will be restricted only to authorized personnel of the research team. All participant data will be hosted by the responsible researcher in protected folders in the cloud of the institutional Microsoft OneDrive server located in the European Economic Area to guarantee privacy.

### 2.10. Statistical Analysis

#### 2.10.1. Univariate Data Analysis

The data obtained from families in this study will be evaluated via an intention-to-treat (ITT) analysis, including data from families who drop out of the study, and per protocol (PP), that is, those families who have completed the treatment plan and who have exactly followed the instructions of the trial protocol. The error level of 0.05 will be set as a limit of statistical significance. A data analysis will be performed using the Statistical Software for Data Science (STATA, College Station, TX, USA) version 17.0 [63].

Mean and standard deviation will be reported for normally distributed variables and quantiles (25th percentile, median, and 75th percentile) for non-normally distributed variables. The Kolmogorov–Smirnov test and the Shapiro–Wilk test will be used to test for the normality of variables. The resulting scores of adherence to the MD (measured from the MEDAS and KIDMED questionnaires) will be used to calculate the mean differences between each intervention group and the control group before (V1) and after interventions (V2). Multiple comparisons (V2 vs. V0 and V2 vs. V1) of mean differences before and after interventions will be undertaken to assess the independent effect of interventions. The normality of the differences will be checked and, depending on the results, the following contrasts will be performed: parametric statistics (three-factor ANOVA with post hoc tests using the Bonferroni correction and considering the interactions between the following factors: app, educational materials, and snack products) will be used for normally distributed variables to compare differences in means and nonparametric statistics (Kruskal–Wallis test) will be used for non-normally distributed variables. Lineal regression and Spearman’s method will be used to test for correlations between normal and non-normal outcome variables, respectively.

#### 2.10.2. Multivariate Data Analysis

Structural equation modeling (SEM) is a multivariate, hypothesis-driven technique that is based on a structural model representing a hypothesis about the causal relations among several variables. It will integrate the multiple factors affecting the adherence to the MD, including economic factors, social factors, psychological factors, quality of life, attitudes and beliefs related to food, health and healthy lifestyle, dietary habits, nutritional knowledge, and physical activity. It will test the direct and indirect effects of these multiple factors on the adherence to the MD. Fit indices like the root mean square error of approximation (RMSEA), the Tucker–Lewis index (TLI), and the comparative fit index (CFI) will be used to check for the adequacy of the fit of the model.

We will monitor model fit indices (RMSEA, TLI, and CFI) to ensure they fall within acceptable ranges and adjust the SEM model structure as necessary based on theoretical and statistical considerations to maintain interpretability. Missing data will be handled using Full Information Maximum Likelihood (FIML) estimation, an approach suited to SEM that reduces bias by using all available data without directly imputing values. Additionally, we will evaluate patterns of missingness to determine if data are missing at random (MAR) or missing not at random (MNAR) and will justify our handling approach accordingly. The reliability will be checked using three criteria, including the internal consistency, composed reliability, and average variance extracted (AVE). Cronbach’s alpha coefficient will be used to test the internal consistency of the scales. For validity, two criteria will be tested, including the convergent validity and the discriminant validity [62].

To address the socioeconomic and cultural differences among participants from Spain, Turkey, and Morocco, a stratified analysis will be conducted to examine the impact of socioeconomic and cultural contexts on participants’ responses to the interventions. By categorizing participants based on key demographic variables such as income, education level, and cultural background, we can gain insights into how these factors influence dietary behaviors and adherence to the MD.

## 3. Discussion

The long-term feasibility of nutrition education programs is limited at the family level, so it has been suggested to use complementary nutrition interventions to build a supportive environment for effective healthy eating in the family setting [28]. To the best of our knowledge, the SWITCHtoHEALTHY study is the first multicentric nutritional intervention study assessing the effectiveness of a multi-component intervention to improve the adherence of Mediterranean families to the MD eating pattern through sustained changes in dietary behavior.

The novelty of the SWITCHtoHEALTHY study relies on the holistic family-based approach that combines and adapts different interactive tools, snack products, and methodologies designed to foster adherence to the MD of the different family members (parents, adolescents, and children) by considering their specific needs. The proposed approach involves the four main driving forces that, together, improve adherence to the MD eating pattern: sociocultural, economic, environmental, and health–nutritional [64].

Despite its well-known benefits, the MD is being abandoned or not adopted by young generations in most Mediterranean countries. The erosion of the MD dietary pattern in the modern era came with globalization, since it is often cheaper to import food from abroad than making local food available. Together with this, the increase in the consumption of ultra-processed food, food insecurity, and youth unemployment are all predisposing factors to unhealthy eating behavior [4,13]. In Spain, up to 69% of the child and adolescent population has been found to have suboptimal adherence to the MD according to the KIDMED index. The prevalence of low adherence is higher for secondary school than for primary school children [65,66]. In Moroccan and Turkish populations, only about 15% of adolescents have optimal adherence to the MD; among the factors associated with greater adherence, female gender, high monthly family income, and living in an apartment stand out [67,68,69]. In the same way, adolescents and young people living in Europe are far from being compliant with the nutritional recommendations for fruit, vegetables, legumes, and sodium, and they do not follow the principles of the MD [70,71]. Some factors positively associated with an optimal adherence to the MD are the mother’s education level, the absence of distractions at breakfast, and regular physical activity, all factors that depend largely on the family environment [66].

Recently, healthy lifestyle-based interventions have been shown to be effective in increasing adherence to the MD and in achieving optimal adherence to this dietary pattern among children and adolescents [25,72]. According to a recent meta-analysis, greater improvements in achieving optimal adherence to the MD are found in interventions delivered out of school, those aimed at both children and parents, and those including participants with overweight/obesity [25]. The SWITCHtoHEALTHY intervention will adopt a “treat the family” approach to strengthen the adherence to the MD, focusing on the families and particularly families with children (kids and adolescents) for two main reasons: firstly, meals are one of the most important social activities among family members and one of the essences of the MD that is currently being abandoned; secondly, children’s food habits are primarily acquired within families, and some of these food habits will persist over time. Families will be the unit of analysis as we try to understand the environmental factors affecting food consumption. Ultimately, this study aims to promote the MD by empowering families to be the actors of such behavioral change, generated from and for the family setting. This is a novelty when compared to other individual-driven interventions focused on specific age targets.

The use of the S2H app will support parents in preparing healthier weekly dietary plans, which will lead to greater adherence to the MD of the entire family. According to prior studies, technology-based interventions delivered via smartphone apps are successful in helping individuals achieve better improvements towards MD adherence in the short term [73] and have shown satisfactory usability, especially among young people [74]. Given the widespread use of digital devices, the potential of smartphone use in low- and middle-income countries has already been highlighted in the literature [75,76]. mHealth has been reported in the literature as a widely used tool to improve health outcomes for vulnerable communities in developing countries worldwide, such as those in Africa, Asia, and Latin America. The importance of implementing these methodologies according to the sociocultural needs of each population group is reiterated in the SWITCHtoHEALTHY project. The app developed within the project is based on the findings of a preliminary phase of research into the habits and needs of each individual country involved in this study. It also addresses this challenge by providing meal plans based on the MD, aligned with the cultural traditions of each participating country.

On the other hand, the development of novel healthy snack products has the potential to improve the contribution of essential nutrients in children’s diets and, in the longer term, may reduce the impact of poor nutrition on public health [77]. Snacks are generally eaten between main meals, often with the intention of reducing or preventing hunger until the next meal. Healthy food options, such as fruits, nuts, and vegetables, should be promoted as between-meal snacks to avoid the consumption of energy-dense and processed foods like chips, biscuits, and sweets [78]. In this context, the SWITCHtoHEALTHY plant-based snacks will be made with a variety of local fruits, vegetables, legumes, and nuts to cover the nutritional needs of children in the mid-morning and mid-afternoon periods. It should be noted that the ingredients used for its preparation will be easily accessible in each country and can even be prepared in a similar way at home by families in the long term, for example, in the form of blends, purees, and bars.

The main strengths of this study are the multicentric nature of the intervention and the transferability and scalability of the educational materials designed in the frame of the project to be reproduced and replicated in other environments and countries beyond the project boundaries (e.g., workplaces, schools, campus canteens, and restaurants). During the intervention, the educational materials will be used at the family level to educate families on the MD and associated health benefits through a toolkit that combines information and tips to cook, prepare foods, benefits, and culture behind MD. In addition, adolescents will conduct educational activities codesigned with them that will reinforce their healthy dietary habits. Once the intervention ends, the materials could be used at the school level, serving as educational resources for teachers to become food educators. In fact, previous studies have shown the effectiveness of nutritional interventions led by trained teachers on improving adherence to the MD in schoolchildren, e.g., The Nutrition Education Teaching Pack (TP) funded by the Italian Ministry of Agricultural, Food, and Forestry Policies [79]. However, the adaptability of the materials is not exempt from challenges, especially when talking about populations outside the Mediterranean region. Materials may need adaptation to account for social and cultural differences. Local language translations would be essential to ensure accessibility and comprehension for participants from different linguistic backgrounds. Finally, resource availability, both in terms of foods typical of the MD and the economic means to procure them, would need to be addressed in any region-specific adaptation, ensuring alignment with local capacities and constraints.

On the other hand, the SWITCHtoHEALTHY study will play a key role in the achievement of a large part of the sustainable development goals (SDGs) linked to EU policies [80]. This will be, to a greater extent, possible through actions such as the transference of knowledge and skills for sustainable development; the valorization of food products from the traditional MD by using local plant varieties and eco-friendly food processing technologies; and the reduction in global food waste and the carbon footprint by promoting a sustainable consumption. In addition, it is expected that the intervention enhances food security and nutrition by providing personalized meal plans, improving diet quality indexes for at least 75% of participants.

Some potential limitations need to be considered. First, given that the dietary habits of children under 8 years of age will be answered by parents, it can suppose an underestimation or overestimation of the consumption of certain MD food groups, so the results could be biased. Second, the differences in socioeconomic and cultural factors among participating families could influence the changes in dietary habits given that certain typical MD foods such as virgin olive oil may not be accessible to families with lower incomes. To account for this, our approach incorporates data collection on household income and food expenditure through structured questionnaires. This will allow us to analyze how financial factors impact dietary choices and adapt our strategies to accommodate families’ economic realities. In addition, the fact that the interventions will take place in three different cultural areas could bias the results given different culinary traditions and beliefs around Mediterranean foods. Despite this, a tailored approach will be used for the design of menus and educational materials adapted to country specificities. Third, due to the nature of the interventions, participants and researchers cannot be blinded, although researchers who will perform the statistical analyses will be. Fourth, the gap in time between V0 and the start of the intervention (V1) can lead to a high drop-out rate of families. Another limitation is the use of the family as a unit of analysis, which means that all members must be involved and willing to lead the intervention to achieve change. Additionally, the short intervention period (3 months) might limit conclusions on the long-term effects. The three-month timeframe was chosen due to the limited duration of the project and the long time needed for the design of the study and the intervention components. This short-term assessment provides a foundation for understanding the intervention’s immediate effectiveness and establishes groundwork for future studies to explore the long-term sustainability of MD adherence. Finally, families could experience a loss of adherence to the intervention components, particularly to the use of the app. The use of eHealth tools presupposes that users have a certain level of skills and competence (eHealth literacy) [81]. In this sense, the S2H app will be user-friendly and universally accessible, and it will have a usability monitoring system.

Regarding the long-term adoption of healthy eating habits at the family level through multi-component interventions such as those proposed in this study, it is worth considering some barriers that may hinder adherence to interventions such as antisocial behavior during adolescence or the lack of parental engagement due to lack of time. For this reason, family-based interventions may focus on the provision of skills, knowledge, and support; frequency and quality of parent–child communications; and reinforcement of shared values and behaviors ensuring the involvement of all family members [17]. The holistic family-based approach planned for the SWITCHtoHEALTHY study will consider all these barriers associated with family behavior. Furthermore, the use of development methodologies such as the learning-through-playing approach will encourage children and adolescents to develop cognitive and communication skills, learn to manage their emotional states, and gain self-confidence for changing their eating behaviors [82].

While this study focuses on promoting adherence to the core components of the MD, such as the increased consumption of fruits, vegetables, whole grains, legumes, and extra virgin olive oil, we acknowledge the recent shifts in dietary habits in Mediterranean countries. The rising intake of ultra-processed foods and the use of oils other than extra virgin olive oil are noteworthy trends that may influence adherence to traditional dietary patterns.

The multidisciplinary consortium will facilitate the exchange of best practices among Mediterranean basin countries to create common knowledge and understanding on the impact of greater adherence to the MD at the health, environmental, and economic level. It will facilitate generating, boosting, and maintaining the switch to a healthier Mediterranean dietary pattern across the Mediterranean area. The key findings generated will be translated into the development of new future strategies to increase the adaptation of individuals to the MD by encouraging nutritionally adequate, healthy, and sustainable behaviors [83]. The results will be summarized, and recommendations will be prepared for government bodies, public health institutions, and NGOs across the Mediterranean. These documents will translate scientific results into actionable policies. In addition, a series of workshops and roundtables will be organized in collaboration with national health ministries and local authorities in Spain, Morocco, and Turkey. The most impactful components (e.g., plant-based snacks and digital tools) will be proposed for inclusion in national and regional health programs, with the support of cost–benefit analyses demonstrating their scalability and sustainability. By employing these mechanisms, SWITCHtoHEALTHY aims to ensure that its findings do not remain confined to the academic sphere but are translated into practical strategies adopted by policymakers and public health institutions across the Mediterranean region.

## Figures and Tables

**Figure 1 nutrients-16-03938-f001:**
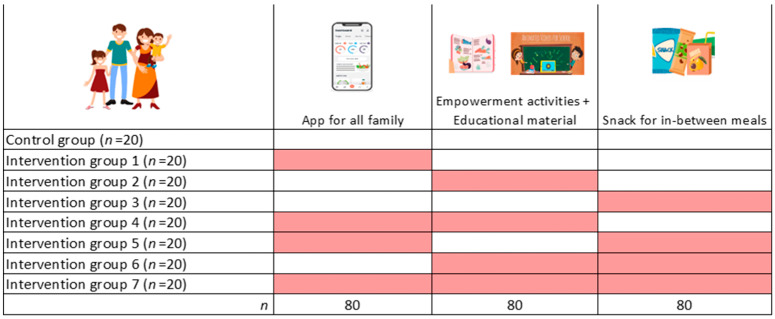
Overview of the study groups.

**Figure 2 nutrients-16-03938-f002:**
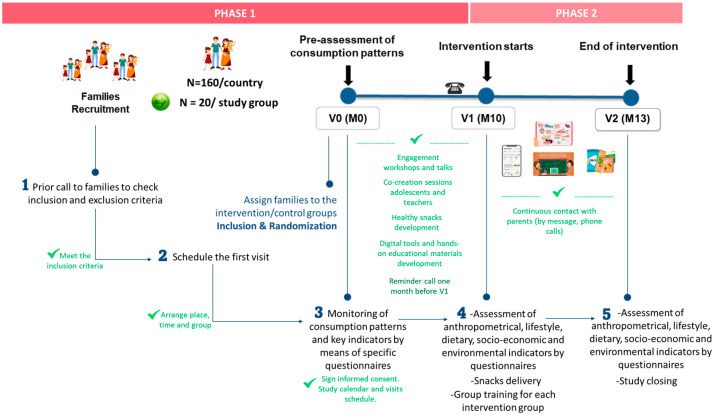
Timeframe and families’ journey along the multicentric nutritional intervention study.

**Figure 3 nutrients-16-03938-f003:**
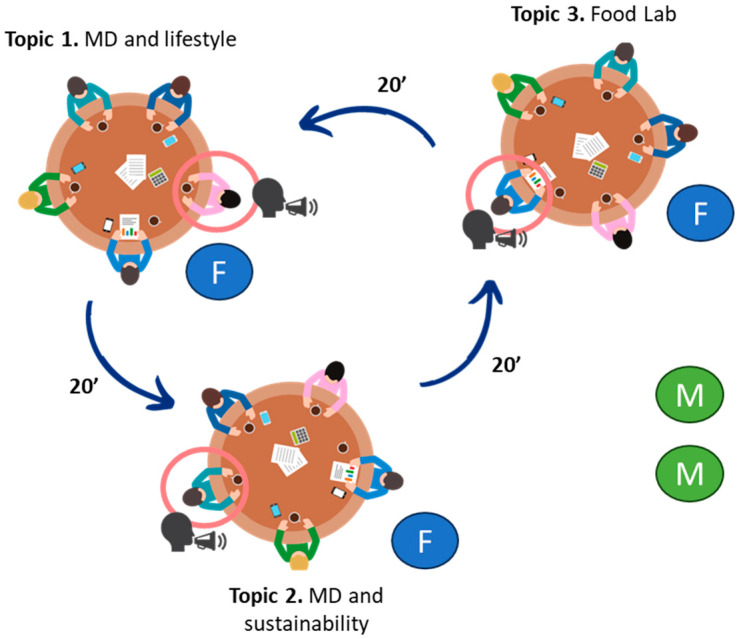
Codesign of educational activities: World Café Methodology.

**Figure 4 nutrients-16-03938-f004:**
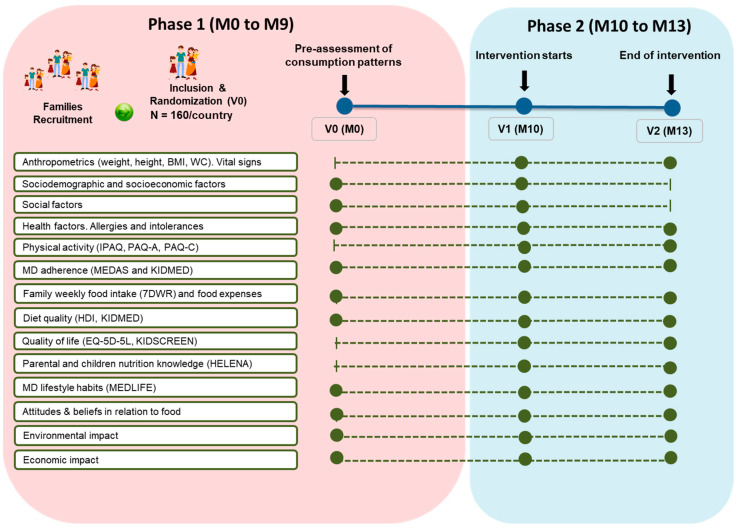
Description of study visits and outcomes.

**Table 1 nutrients-16-03938-t001:** Components of the S2H app.

App Component	Description
AI-Based App for Parents	The app will be accessible by parents from any device (including Android/iOS smartphones) through a web browser. The users will be able to provide basic profile information and receive personalized weekly meal plans adhering to a healthy and sustainable Mediterranean-based diet. The app will co-ordinate with the app for children’s Med-based dietary plan to provide personalized weekly meal plans for all family members (parents, adolescents, and children).
App for Children’s Med-Based Dietary Plan	The menus for children will complement the local school catering services (for nursery, kindergarten, and primary schools) and provide parents with ideas and suggestions about breakfast, snack, dinner, and weekend preparation, promoting a balanced food intake during the week. The app will thus consider what children eat in school canteens (based on the details provided by the school catering system), in the school café, and in what they bring in their lunch boxes from home (details provided by parents) and calculate possible complementary options for the rest of the day at home and for the weekend. In addition, educational materials, recipes, and daily tips are also provided to improve parents’ culinary skills and nutrition knowledge, empowering them to manage the whole family’s diet and to make proper food choices.
Digitami Game for All the Family	Digitami is an educational life simulation game to increase family engagement. It is envisioned as a “slow-paced” mobile game where the family members will have to take care of a “Tamagotchi” digital character living in the app, choosing and preparing their daily meals and/or physical activities to follow a balanced diet and a healthy lifestyle. Simultaneously, it will provide feedback about the nutritional value of the player’s choices. Push notifications will be used to increase the engagement level of the family members.

**Table 2 nutrients-16-03938-t002:** List of ingredients presented to the families in the focus groups for each country.

Country	List of Selected Ingredients			
	Vegetables	Fruits	Dried Fruits and Nuts	Legumes and Cereals
Spain	Pumpkin	Mandarin	Almond	Chickpea
		Grape		
		Peach		
	Spinach	Nopal		Rice
			Walnut	Grains
	Sweet potato	Sumac		
			Hazelnut	
Turkey	Chicory	Jujube	Dates	Chickpea powder
			Dried apricot	
			Almond	
			Walnut	
		Black fig		
				Oatmeal
			Hazelnut	
		Black mulberry		
		Blueberry		
	Black carrot			
Morocco	Beetroot	Fig	Dates	Chickpea
				Whole wheat
		Orange		
		Clementine		
		Banana		
			Almond	
				Oat
			Peanut	
		Apricot	Tigernut	
		Apple		
		Carob		

**Table 3 nutrients-16-03938-t003:** Blocks of questions in the online forms for families.

Block	Assessed Factors
Socioeconomic factors	Age, gender, ethnicity, household members, marital status, income, level of education, and employment status
Social factors	Work flexibility and satisfaction, work–life balance, lifestyle, time spent with children, family decision making, and task responsibility
Psychological factors	Self-perception
Quality of life	Mobility, self-care, usual activities, pain/discomfort, anxiety/depression, and subjective well-being
Attitudes and beliefs related to food	Family eating habits and attitudes of and barriers faced by parents and adolescents related to the MD
Health and healthy lifestyle	Smoking, hospitalization, medication use, and sleep duration
Dietary habits	Eating rules, amount and type of food consumed, and food frequency intake
Nutritional knowledge	Level of knowledge of parents and adolescents on a healthy diet
Physical activity	Types and intensity of physical activity and sitting time spent as part of the individual’s daily life

## Data Availability

Data will be made available upon reasonable request.

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
