# Peer review of "Switching Mediterranean Consumers to Mediterranean Sustainable Healthy Dietary Patterns (SWITCHtoHEALTHY): Study Protocol of a Multicentric and Multi-Cultural Family-Based Nutritional Intervention Study"

_nutrients, 2024, doi:10.3390/nu16223938_

Round 1
Reviewer 1 Report
Comments and Suggestions for Authors
Abstract:
1. The abstract effectively highlights the problem of the Mediterranean population abandoning the traditional Mediterranean Diet (MD) due to socio-economic changes. This provides a strong justification for the intervention.
2. Using a multi-component intervention (digital tools, educational materials, snacks) targeting multiple aspects of behaviour is promising. Addressing nutrition and engagement through tools like a life simulation game shows innovation in intervention design.
3. Including families from three Mediterranean countries (Spain, Morocco, and Turkey) adds cultural diversity to the study, improving the findings’ generalizability across the Mediterranean region.
4. The primary outcome (adherence to the MD, measured via MEDAS in adults and KIDMED in children) is clearly defined and relevant to the research goals.
5. The authors mention “profound socio-economic changes” but do not specify these changes. Clarifying these would help better understand the context and necessity of the intervention.
6. The rationale behind selecting specific components (e.g., plant-based snacks and digital tools) must be explained better. Why are these the most effective methods for improving MD adherence?
7. A three-month intervention might not be sufficient to induce long-term dietary changes, especially in the context of family habits. The abstract doesn't mention any follow-up to assess the intervention's sustainability. Long-term adherence could be critical to understanding the intervention's actual effectiveness.
8. The theoretical underpinnings guiding the intervention should be mentioned. Behavioural change interventions often rely on behaviour change models, but none are mentioned here. Including a theoretical framework could strengthen the intervention's design.
9. While it is mentioned that the intervention’s impact on diet quality, economy, and the environment will be assessed, there is no mention of how these will be measured or why they are relevant to the study's goals. Including more detail about the methods for assessing these factors would add clarity.
Introduction
1. While the list of participating institutions and international partners adds credibility to the project, it feels overly detailed for an introduction. This section could be condensed or moved to another part of the manuscript (e.g., Materials and Methods) to maintain focus on the research aims.
2. Although the introduction mentions the general need for family-based dietary programs, it doesn’t explicitly highlight specific gaps in current interventions or why the existing methods have been insufficient. Addressing these would help better justify the need for the SWITCHtoHEALTHY project.
3. The introduction needs to include a theoretical framework guiding the intervention. Mentioning the rationale for why this particular approach (family-based, digital tools, etc.) is expected to be effective is needed.
4. While school-based nutrition programs are mentioned, their relevance to SWITCHtoHEALTHY's family-based focus needs to be clarified. A more direct comparison to family-based interventions would strengthen the argument.
5. The authors need to acknowledge potential challenges or limitations of the intervention (e.g., adherence to digital tools and cultural barriers). Discussing these briefly would make the proposal more realistic and thoughtful.
6. Specific points, such as the description of the MD and the importance of family influence, are repeated multiple times in different contexts, which could be more concise.
7. The authors briefly mention using mobile Apps to promote MD but don’t explain the potential strengths or challenges of using digital tools as part of the intervention. Elaborating on how digital tools can effectively change dietary behaviour, especially in a family setting, would strengthen the rationale.
Materials and Methods
- While the study is single-blinded, it leaves room for potential observer bias among researchers aware of the groups. Consideration should be given to a double-blind design, where participants and researchers involved in the data collection are blinded.
- Although the multicentric design enhances generalizability, the socioeconomic and cultural differences between Spain, Turkey, and Morocco may introduce variability in how participants respond to the interventions. The authors should address how these factors will be accounted for in the analysis.
- The intervention period is only three months. While this is sufficient for testing short-term changes in dietary patterns, it may need to be longer to evaluate long-term adherence to the Mediterranean diet or sustainable lifestyle changes, especially for children and adolescents.
- The inclusion criterion that families must have both a child younger than 12 years and an adolescent may create an inconsistent family structure across groups, affecting intervention implementation.
- The exclusion criteria rule out families that already adhere to the Mediterranean Diet (KIDMED score ≥ 11). While this helps focus on families who need intervention, it narrows the population, potentially overlooking differences in effect for families already somewhat aligned with MD principles but not at an optimal level.
Outcomes
1. The binary scoring system (0 or 1) may need to be more balanced with the complexity of dietary adherence. For instance, a participant who eats a food occasionally might score the same as someone who never eats it. A more nuanced scoring system (e.g., frequency-based) could yield more accurate insights.
2. The extent to which these questionnaires account for cross-cultural differences in MD interpretation needs to be fully clarified. Although the MD is universal, its application might vary between countries, which could affect results in a study across multiple regions.
3. Although the Tanita scale and stadiometer are reliable, their accuracy can vary depending on the conditions during measurements (e.g. time of day, etc.). Variability in measurement protocols across countries could introduce biases.
4. Systolic and diastolic blood pressure are acute measures, and while important, changes might not be significant over the short duration of the intervention. Longitudinal measurements would provide more meaningful insights into the MD's cardiovascular benefits.
5. While the questionnaire gathers essential data, socioeconomic factors are complex and dynamic. Self-reported income and job status might not fully capture a household’s financial stability or stress levels, which can influence dietary choices.
6. Physical activity questionnaires are prone to reporting biases, as people tend to overestimate their physical activity. Objective measures (e.g., pedometers or accelerometers) could provide more accurate data.
7. There might be some overlap between what’s being measured in the GHQ, MEDLIFE, and MEDAS scores, which could lead to redundancy in data collection without providing much new insight.
8. Quality of life is a highly subjective measure. It can be influenced by many factors outside the scope of the study, such as recent life events that are not directly related to MD adherence.
9. The process of weighing food and collecting receipts could be burdensome for participants, potentially leading to lower compliance or inaccurate data if not consistently followed.
10. Converting food intake into environmental metrics is complex and prone to error if not done precisely, especially across different food types and regions.
11. A 25% dropout rate is assumed, but the burden of participation (food tracking, receipt collection, etc.) could lead to a higher dropout rate, which may affect study outcomes and bias results.
12. The SEM approach, while powerful, could introduce difficulties in interpretation, especially if the model fit indices (RMSEA, TLI, CFI) are outside acceptable ranges. Additionally, handling missing data or dropouts in the analysis should be well-justified.
Discussion
1. The authors commendably highlight the novelty of the family-based approach in the SWITCHtoHEALTHY study. Involving parents, adolescents, and children while considering their specific needs adds credibility to the intervention's effectiveness. However, while the multidimensional approach (socio-cultural, economic, environmental, and health-nutritional) is comprehensive, the paper could have provided more detail on how each domain will be operationalized in practice. For instance, how will socio-economic constraints be accounted for beyond simply acknowledging them as limitations?
2. The authors should investigate why these young generations are moving away from the MD. Factors such as globalization, exposure to Westernized diets, and changing lifestyles should be explored more explicitly.
3. The discussion could benefit from addressing potential limitations of prior interventions, such as cultural resistance or difficulty sustaining dietary behaviour changes over the long term.
4. Reliance on technology raises concerns about accessibility for families with lower technological literacy or limited smartphone access. The discussion should more thoroughly address potential digital divides.
5. The long-term effects of relying on snack products rather than whole foods to promote adherence to the MD deserve more exploration in the discussion.
6. While the authors mention the materials’ adaptability, they can offer more insight into the potential challenges of transferring them across different cultural and economic contexts outside the Mediterranean region.
7. One limitation that requires further emphasis is the use of the family as the unit of analysis. The intervention relies heavily on the willingness of all family members to participate, which may only be feasible in some households.
8. In addition, the short intervention period limits conclusions about long-term behavioural change, which deserves further elaboration regarding how the study will address this gap in future research.
9. Outlining the specific SDGs targeted and how the SWITCHtoHEALTHY study will contribute quantitatively to those goals could benefit the discussion.
10. The study’s potential to generate new strategies for promoting MD across Mediterranean countries is clear. However, the authors should clarify the mechanisms by which the study’s results will be disseminated and adopted by policymakers and public health institutions.
Comments on the Quality of English LanguageThe English can be improved
Reviewer 2 Report
Comments and Suggestions for Authors
The research proposal is very interesting and well designed. Unfavorable dietary changes are generating various nutritional problems. In this regard, this proposal could provide very valuable information. I only have the following (minor) comments.
I. Comments.
1. I suggest including more specific aspects about dietary changes that are occurring in Mediterranean countries, for example, intake of other oils (different from extra virgin olive oil) or ultra-processed foods.
2. Considering the proposed methodology, will researchers be able to obtain information by age group? Considering especially children and adolescents?
3. In addition to the diet as a general pattern, it would be very interesting if the authors manage to obtain information about the specific intake of foods and nutrients.
